# Optimizing fungal DNA extraction and purification for Oxford Nanopore untargeted shotgun metagenomic sequencing from simulated hemoculture specimens

Nattapong Langsiri,[1] Wieland Meyer,[2,3] Laszlo Irinyi,[3] Navaporn Worasilchai,[4,5] Nuttapon Pombubpa,[6,7] Thidathip Wongsurawat,[8,9] Piroon Jenjaroenpun,[8,9] J. Jennifer Luangsa-ard,[10] Ariya Chindamporn[1,11]

**ABSTRACT**   Long-read metagenomics provides a promising alternative approach to fungal identification, circumventing methodological biases, associated with DNA amplification, which is a prerequisite for DNA barcoding/metabarcoding based on the primary fungal DNA barcode (Internal Transcribed Spacer (ITS) region). However, DNA extraction for long-read sequencing-based fungal identification poses a significant challenge, as obtaining long and intact fungal DNA is imperative. Comparing different lysis methods showed that chemical lysis with CTAB/SDS generated DNA from pure fungal cultures with high yields (ranging from 11.20 ± 0.17 µg to 22.99 ± 2.22 µg depending on the species) while preserving integrity. Evaluating the efficacy of human DNA depletion protocols demonstrated an 88.73% reduction in human reads and a 99.53% increase in fungal reads compared to the untreated yeast-spiked human blood control. Evaluation of the developed DNA extraction protocol on simulated clinical hemocultures revealed that the obtained DNA sequences exceed 10 kb in length, enabling a highly efficient sequencing run with over 80% active pores. The quality of the DNA, as indicated by the 260/280 and 260/230 ratios obtained from NanoDrop spectrophotometer readings, exceeded 1.8 and 2.0, respectively. This demonstrated the great potential of the herein optimized protocol to extract high-quality fungal DNA from clinical specimens enabling long-read metagenomics sequencing.

**IMPORTANCE**   A novel streamlined DNA extraction protocol was developed to efficiently isolate high molecular weight fungal DNA from hemoculture samples, which is crucial for long-read sequencing applications. By eliminating the need for labor-intensive and shear-force-inducing steps, such as liquid nitrogen grinding or bead beating, the protocol is more user-friendly and better suited for clinical laboratory settings. The automation of cleanup and extraction steps further shortens the overall turnaround time to under 6 hours. Although not specifically designed for ultra-long DNA extraction, this protocol effectively supports fungal identification through Oxford Nanopore Technology (ONT) sequencing. It yields high molecular weight DNA, resulting in longer sequence fragments that improve the number of fungal reads over human reads. Future improvements, including adaptive sampling technology, could further simplify the process by reducing the need for human DNA depletion, paving the way for more automated, bioinformatics-driven workflows.

**KEYWORDS**   DNA preparation, long read, simulated hemoculture, Oxford Nanopore Technology

The emergence of invasive fungal infections presents significant challenges in clinical mycology, primarily due to the absence of rapid and sensitive diagnostic methods

Address correspondence to Ariya Chindamporn, drariya@gmail.com.

The authors declare no conflict of interest.

See the funding table on p. 18.

*[This article was published on 8 April 2025 with errors in Table 1 and Fig. 8. The errors were corrected in the current version, posted on 14 May 2025.]*

(1). Conventional culture-based diagnostics are slow, leading to delayed diagnoses that can negatively impact patient survival (2–5). As such, faster and more precise molecular diagnostics are essential. While DNA barcoding targeting the ITS region is preferred, it is also challenging due to the complexity of the fungal kingdom and the high variability in the ITS sequences (6). The absence of standardized protocols for specimen handling, DNA extraction, PCR method, and result interpretation complicates the use of PCR-based diagnostics for invasive fungal infections (7, 8).

Untargeted metagenomic sequencing holds promise for pathogen identification by avoiding biases associated with limited target markers and the need for DNA amplification, thereby reducing false negatives and enabling more accurate fungal identification (9–11). However, next-generation sequencing produces short sequences (~50–300 bp) (12) that require fragment merging, potentially introducing errors (13). Long running times, often over 12 hours, hinder real-time clinical analysis. Sequencing frequently prioritizes human DNA, resulting in a poor pathogen coverage or misidentification (13, 14). The high volume of generated reads also poses management challenges for individuals without bioinformatics expertise.

In contrast, ONT long-read sequencing enhances fungal identification by generating longer sequences, allowing for more accurate classification at both the species and genus levels. However, obtaining high-quality DNA, which is essential for long-read sequencing, from fungi is challenging due to the complex and diverse fungal cell wall structures (15, 16) and components (17–19), making it hard to find a universal lysis method that avoids significant DNA shearing (20). This impacts DNA integrity and purity, both of which are essential for successful sequencing (21). Intact DNA is crucial for long-read sequencing, as it enables the generation of longer sequence reads, which improves taxonomical classification in untargeted metagenomic sequencing (22). ONT recommends DNA purity criteria with 260/280 ratios between 1.8–2.0 and 260/230 ratios ranging from 2.0 to 2.2 (23).

Physical lysis methods like bead beating and liquid nitrogen grinding can yield high DNA amounts but carry the risk of contamination (20, 24) and degradation due to the generated heat during the process (17). Enzymatic lysis with zymolyase preserves DNA integrity (25) but does not work for all fungal species (26) and can introduce contamination (27). Chemical lysis, using detergents like sodium dodecyl sulfate (SDS) or cetyltrimethylammonium bromide (CTAB), disrupts cell structures to isolate DNA (28) but may introduce contaminants (29). CTAB, in particular, helps separate polysaccharides from DNA, thereby improving purity (30). Combining CTAB, with SDS and using ultracentrifugation eliminates the need for toxic phenol/chloroform extraction and increases DNA yield, though it can still impact long-read sequencing technologies like ONT due to potential contamination and pore inactivation (19, 31). However, there are evidences that support the use of both chemical agents (CTAB and SDS), which increase yield while maintaining the integrity of the DNA in polysaccharide-rich organisms, such as plants or fungi (32–34).

In addition to the extraction steps, depleting host DNA is essential for untargeted shotgun metagenome sequencing (35, 36). The presence of irrelevant DNA, such as host DNA in clinical samples, can disrupt and interfere with downstream analysis (8, 9, 13). ONT's new adaptive sampling feature has the potential to transform this process. It allows users to upload reference DNA sequences to selectively deplete or enrich specific DNA during sequencing (37). However, this innovation is exclusive to ONT. Adaptive sampling may also introduce errors by accepting or rejecting reads based on the database used, potentially leading to false classifications or reduced test sensitivity (38, 39). An alternative approach to depleting human DNA is wet lab preprocessing, such as the saponin-based differential lysis method (40). This method has been proven to be effective when applied to respiratory specimens [endotracheal aspirate, sputum, bronchoalveolar lavage fluid (BALF)] (1, 40–42), cerebrospinal fluid (CSF) (35, 43), other body fluid (44), tissue (45), or blood (46, 47). However, its implementation remains limited in fungi (48). Therefore, the success of adaptive sampling or wet lab preprocessing

largely depends on understanding the specific characteristics of each specimen (48, 49). Currently, no universal approach can be effectively applied to all specimens.

Herein, we present a simplified method for extracting high molecular weight fungal DNA from blood specimens, eliminating the need for liquid nitrogen grinding and phenol/chloroform extraction steps. This approach uses a CTAB/SDS lysis buffer followed by automated magnetic bead clean-up. This method, tested in combination with the MinION sequencer, provides a safe and efficient in-house DNA extraction protocol for long-read sequencing of DNA extracted from both pure cultures and clinical samples.

## MATERIALS AND METHODS

### Clinical strain preparation

Six fungal isolates from various clinical specimens were included in this study (Table S1). *Nakaseomyces glabratus* (formerly *Candida glabrata*) (CUCGL01), *Cryptococcus neoformans* (CUCNE41), *Aspergillus fumigatus* (CUAFU24), *Sporothrix schenckii* (CUSSC26), *Rhizopus microsporus* (CURMI38), and *Pythium insidiosum* (CUPIN58). These isolates represent a range of fungal morphologies associated with systemic infections, including easy- and hard-to-lyse yeasts, filamentous molds, dimorphic fungi, and mold-like organisms, which are commonly found in systemic infections. Each isolate was identified by ITS sequencing using the primers ITS1 (5′ TCCGTAGGTGAACCTGCGG 3′) and ITS4 (5′ TCCTCCGCTTATTGATATGC 3′) (6). All isolates were sub-cultured in 100 mL of Sabouraud Dextrose Broth either for 48 h (for *N. glabratus*, *R. microsporus*, and *P. insidiosum*), 72 h (for *A. fumigatus* and *C. neoformans*), or 7 days for *S. schenckii*, followed by agitation at 120 rpm at 37°C. The metadata of the studied isolates are listed in Table S1. Subsequently, we evaluated the optimal DNA extraction method using spiked blood samples, with a human DNA depletion protocol to assess its compatibility with the DNA isolation method (spin column vs paramagnetic beads). Finally, the simulated hemoculture spiked with a fungal culture was tested to evaluate the final extraction protocol for all studied fungal species.

### Physical lysis testing from pure culture

Ten milligram samples from each isolate were subjected to physical lysis using two methods, bead beating and grinding in liquid nitrogen, as described in Table 1. Both methods (Methods 1 and 2 from physical lysis in Table 1) underwent lysis with 700 µL buffer containing Proteinase K (cat. no. 19133, Qiagen, Maryland, USA), followed by a 3 hour incubation at 65°C prior to the physical lysis. After lysis, phenol/chloroform extraction was used to purify the DNA. DNA concentration and quality were evaluated using a NanoDrop spectrophotometer and a Quantus fluorometer. The DNA integrity was checked by running on a 0.8% agarose gel electrophoresis using a 10 kb DNA ladder (cat. no. DM3100, SMOBIO, Taiwan).

To check the efficiency of the DNA isolation methods for sequencing, DNA from *N. glabratus* (representing an easy-to-lyse yeast) was pooled from both lysis methods and prepared for long-read sequencing using ONT's MinION. The library preparation steps were performed as follows. The extracted DNA was adjusted to 100 fmol (measured with Quantus fluorometer). The extracted DNA was first modified using NEBNext Ultra II End repair/dA-tailing Module reagents (cat.no. E7546, New England Biolabs, Ipswich, UK) and NEBNext FFPE Repair Mix (cat. no. M6630, New England Biolabs, Ipswich, UK). Then, the DNA was cleaned using 1× volume of Agencourt AMPure XP beads (cat. no. A63881, Beckman Coulter, Indiana, USA) and adjust the concentration to 100 fmol (measured with Quantus fluorometer). This was followed by tagging barcodes from Native Barcoding Expansion 1–12 (cat. no. EXP-NBD104, ONT, Didcot, UK) using Blunt/TA Ligase Master Mix (cat. no. M0367L, New England Biolabs, Ipswich, UK). Barcoded DNA samples were pooled equally, each at a concentration of 100 fmol. After that, adapter ligation was performed using Quick Ligation Module (cat. no. E6056, New England Biolabs, Ipswich,

**TABLE 1** The summary of lysis and DNA extraction methods testing to select the optimal lysis method[a]

| Method | Lysis approach | Lysis process | | DNA extraction process |
| --- | --- | --- | --- | --- |
| | | Procedure | Buffer components | |
| 1 | Physical lysis | Bead beating (using 0.5 mm sterile glass beads with 5 min beating) (Mini bead beater, Biospec, Bartlesville, OK, USA) | 0.2 M NaCl, 0.02 M EDTA, 0.04 M Tris, 10% (wt/vol) SDS, 100 µg/mL proteinase K | • Phenol:chloroform:isoamyl alcohol (25:24:1) (DNA isolation) <br> • Centrifugation (13,709 × g, 10 min, 4°C) <br> • 1:1 isopropanol at −20°C 1 h, 2× ethanol wash (DNA precipitation) |
| 2 | | Liquid nitrogen grinding (5 min grinding) | | • Centrifugation (13,709 × g, 10 min, 4°C) <br> • 70 µL elution volume with water (DNA elution) |
| 3 | Chemical lysis | SDS | 10% (wt/vol) SDS, 0.2 M NaCl, 0.02 M EDTA, 0.04 M Tris-HCl | • Ultracentrifugation (18,659 × g, 15 min, 4°C) (DNA isolation) |
| 4 | | Method 3 + proteinase K | Method 3 + 100 µg/mL proteinase K | • 1:1 isopropanol at −20°C 1 h, 2× ethanol wash (DNA precipitation) |
| 5 | | CTAB + proteinase K | 2% CTAB, 1% PVP40, 3% (wt/vol) NaCl, 0.02 M EDTA, 0.04 M Tris-HCl | • Centrifugation (13,709 × g, 10 min, 4°C) |
| 6 | | SDS + CTAB + proteinase K (method 4 + 5 + proteinase K) | 10% SDS, 2% CTAB, 1% PVP40, 3% (wt/vol) NaCl, 0.02 M EDTA, 0.04 M Tris-HCl, 100 µg/mL proteinase K | • 70 µL elution volume with water (DNA elution) |
| 7 | | Method 6 + RNase A | Method 6 + 100 µg/mL RNase A | Method 6 + <br> • RNase A treatment post-DNA extraction, salting out with 3 M NaCl, alcohol precipitation, resuspension |

[a]The optimal extraction method is highlighted in grey.

UK) to ligate the Adapter mix (AMII) from the Native Barcoding Expansion 1–12 (cat. no. EXP-NBD104, ONT, Didcot, UK) to the DNA sample. The library was cleaned up with 0.5× volume of Agencourt AMPure XP beads (following the manufacturer's protocol). Finally, 15 µL of the eluate containing the DNA library was transferred into a clean 1.5 mL tube and adjusted to the concentration of 50 fmol (measured with Quantus fluorometer) for flow cell loading following the manufacturer's protocol. The flow cell was primed with Flush buffer and Flush tether from the Ligation Sequencing Kit 109 (SQK-LSK109, ONT) according to the manufacturer's instructions. The library was mixed with Sequencing Buffer and Loading Beads to a final volume of 75 µL and added dropwise to the SpotON sample port. Basecalling and demultiplexing were performed using Guppy (version 5.0.16, ONT) with the super accuracy base calling model (the r9.4.1_450bps_sup_model provided by ONT), and data quality was assessed with Nanoplot (50).

## Chemical lysis testing on pure culture

Five chemical lysis methods were tested on six fungal isolates to minimize DNA shearing and improve integrity as described in chemical lysis in Table 1. The methods included SDS lysis buffer (Method 3), SDS with Proteinase K (Method 4), CTAB with Proteinase K (Method 5), an optimized combination of SDS/CTAB-PVP40 with Proteinase K (Method 6), and a variation of Method 6 with added RNase A (cat. no. 19101, Qiagen, Maryland, USA) (Method 7). To disrupt the fungal mycelia, initial homogenization was performed by either using Vortex mixer (cat. no. TT-254-VM, Hercuvan Lab Systems, London, UK) with glass beads (cat. no. 11079105, BioSpec, Vernon Hills, USA) or with zirconia beads (cat. no. NC0362415, Biospec, Vernon Hills, USA) for 20 minutes. The DNA was then cleaned via ultracentrifugation (18,659 g for 15 min at 4°C, model: Hettich Universal 320, rotor 1420-B) and precipitated with the addition of 70% alcohol as described in Table 1. The integrity of the extracted DNA was evaluated using 0.8% agarose gel electrophoresis, with the optimal method chosen based on the highest yield, purity, and lowest shearing measured as described in the previous section. Microscopic examination of *N. glabratus* and *A. fumigatus* was conducted using differential interference contrast microscopy (Axioscope 5, ZEISS, Germany) with the lysis methods of Method 4 and 6 over time to

observe the structural changes of the fungal cells at 30 minutes, 1 hour, 2 hours, and 3 hours after lysis buffer treatment.

## Comparison between spin-column DNA extraction and mobilized paramagnetic beads DNA extraction from pure culture

DNA can be extracted from lysates through ultracentrifugation and ethanol precipitation but contaminants with densities similar to nucleic acids might remain in the final DNA elution (32). To reduce excess contaminants, silica-based chromatography was used, which binds and releases DNA based on ionic strength.

To evaluate additional clean-up steps and improve the quality and integrity of DNA, 10 mg of *N. glabratus* and *A. fumigatus* were processed using the lysis Method 6 (a combination of SDS/CTAB with Proteinase K) and tested for DNA quality. The lysate was processed using two silica-based clean-up methods: Maxwell RSC Whole Blood DNA Kit (cat. no. AS1520 Promega, Wisconsin, USA) with paramagnetic beads and Quick-DNA Fungal/Bacterial Miniprep Kit (cat. no. D6005, Zymo Research, California, USA) with a spin column. Both methods were carried out according to the manufacturer's instructions. The optimized lysis and DNA extraction without silica-based clean-up served as a control. The final method was selected based on yield, purity, and integrity measured as described in the previous section. This, combined with a DNA extraction kit (see above), created a faster and easier DNA extraction method (51, 52).

## Human DNA depletion

Removing host DNA is essential for enriching pathogen DNA. The study's protocol was adapted from a previously published method using saponin differential lysis combined with HL-SAN DNase (40). This protocol was modified by adding a pre-treatment step, using 10 mL of eBioscience 1 × RBC Lysis Buffer (cat. no. 00-4333, Invitrogen, Waltham, USA) per 1 mL of human blood before following the published protocol.

A simulated clinical specimen was used to evaluate the compatibility of the human DNA depletion protocol with the different optimized DNA isolation methods. We applied two approaches: paramagnetic beads and spin-column extraction. The concentration of $10^7$ and $10^8$ cells/mL of *N. glabratus*, respectively, were spiked into 1 mL of EDTA blood to mimic blood specimens (53–55) to represent the concentrations of *N. glabratus* positive blood cultures starting with the initial concentration as described previously (55). After depleting human DNA, the pellet was washed, lysed, and the resulting DNA was analyzed for quantity, quality, and integrity as described above.

## Testing the efficacy of human DNA depletion in simulated hemoculture samples

To test the human DNA depletion protocol, four conditions were evaluated: (i) a hemoculture specimen alone, (ii), with 5 mL of human blood applying the depletion protocol, (iii) with 5 mL of human blood without applying the depletion protocol, and (iv) a sample spiked with $10^3$ *N. glabratus* cells/mL. [This spiked concentration provided the optimal sensitivity for conventional and molecular detection with BD BACTEC Myco/F Lytic culture bottles (cat. no. 442288, BD, Maryland, USA) (47, 56–58).] After a positive signal was detected in blood culture bottle, the DNA was extracted and sequenced using Oxford Nanopore's MinION. Reads were filtered by length (1,000 bp, 5,000 bp, 10,000 bp, and unfiltered) using Filtlong 0.2.1 (https://github.com/rrwick/Filtlong) and then classified as human or fungal using BLAST (59) against the human and *N. glabratus* genomes (*Homo sapiens*) (GRCh38.p14 downloaded on 26 August 2022 from https://www.ncbi.nlm.nih.gov/datasets/genome/GCF_000001405.40) and *N. glabratus* (ASM254v2 downloaded on 26 August 2022 from https://www.ncbi.nlm.nih.gov/datasets/genome/GCF_000002545.3).

## Testing the optimized extraction protocol on a simulated hemoculture samples

The optimized protocol for isolating high molecular weight DNA was tested on simulated hemoculture specimens, each spiked with one of the six clinical fungal isolates: *N. glabratus*, *C. neoformans*, *S. schenckii*, *P. insidiosum*, *R. microsporus,* and *A. fumigatus* spores, yeast cells, and zoospores [prepared as described (60)] at a concentration of 200 cells/mL or 200 spores/mL were spiked into 5 mL of blood in BACTEC Myco/F Lytic culture bottles. After incubation in the hemoculture automated system (BD, Maryland, USA) following the manufacturer's protocol until a positive signal was detected, a 1 mL sample from each bottle was subjected to the optimized DNA extraction and human DNA depletion protocol. DNA integrity was assessed by sequencing with ONT's MinION, filtering reads longer than 1,000 bp using Filtlong ver. 0.2.1 (https://github.com/rrwick/Filtlong) and identifying human and fungal reads with BLAST (59) against the reads from the genomes of *H. sapiens* (GRCh38.p14 downloaded on 26 August 2022 from https://www.ncbi.nlm.nih.gov/datasets/genome/GCF_000001405.40), *N. glabratus* (ASM254v2 downloaded on 26 August 2022 from https://www.ncbi.nlm.nih.gov/datasets/genome/GCF_000002545.3), *C. neoformans* (ASM9104v1 downloaded on 26 August 2022 from https://www.ncbi.nlm.nih.gov/datasets/genome/GCF_000091045.1/), *S. schenckii* (S_schenckii_v1 downloaded on 26 August 2022 from https://www.ncbi.nlm.nih.gov/datasets/genome/GCF_000961545.1/), *P. insidiosum* (ASM2283683v1 downloaded on 26 August 2022 from https://www.ncbi.nlm.nih.gov/datasets/genome/GCA_022836835.1/), *R. microsporus* (Rhimi1_1 downloaded on 26 August 2022 https://www.ncbi.nlm.nih.gov/datasets/genome/GCF_002708625.1/), and *A. fumigatus* (ASM265v1 downloaded on 26 August 2022 from https://www.ncbi.nlm.nih.gov/datasets/genome/GCF_000002655.1/). The mean read length and quality were determined and visualized by Nanoplot (50). The pore status and run efficacy were determined with MinKNOW. Moreover, the *de novo* assembly of the metagenomes was performed using Flye (version 2.9.3) (61). The sequence fragments obtained from each strain after assembly were identified to the species level using BLAST (59) against their respective genomes.

## Statistical analysis

Statistical analyses were undertaken using Prism software (GraphPad Software, La Jolla, CA). In all analysis conditions, a *P*-value < 0.05 was considered statistically significant (Confidence interval of 95%). A two-way ANOVA test was used to compare the means of fungal DNA amounts of each extraction method, measured using NanoDrop spectrophotometer and Quantus fluorometer. A one-way ANOVA test was used to compare the mean fungal DNA quantities across the different extraction methods. The Bonferroni correction was applied in all cases as the post-hoc analysis to assess the significance of the mean comparisons.

## RESULTS

### Physical lysis results in extensive shearing of DNA

DNA measurement with Quantus fluorometer indicated poorer DNA quality than that with NanoDrop spectrophotometer (*P*-value < 0.01, 95% CI, Bonferroni-corrected), suggesting the presence of significant contaminants (Fig. 1A). The 260/280 ratio mostly ranged between 1.8 and 2.0 (red line, Fig. 1A), with some isolates slightly lower. In contrast, the 260/230 ratio was generally below the acceptable >2.0, indicating the presence of additional components absorbing at 230 nm (Fig. 1A). After running on 0.8% agarose gel electrophoresis, the result showed moderately sheared DNA from both lysis methods (Fig. 1B). The DNA lengths were further analyzed and visualized using ONT's long-read sequencing platform and Nanoplot. The mean DNA length obtained after extraction from the *N. glabratus* had an $N_{50}$ value of 579 bp for the bead beating method and 886 bp for the liquid nitrogen method (Fig. 1C), consistent with DNA band visualization. MinION sequencing showed that both lysis methods started with over 10%

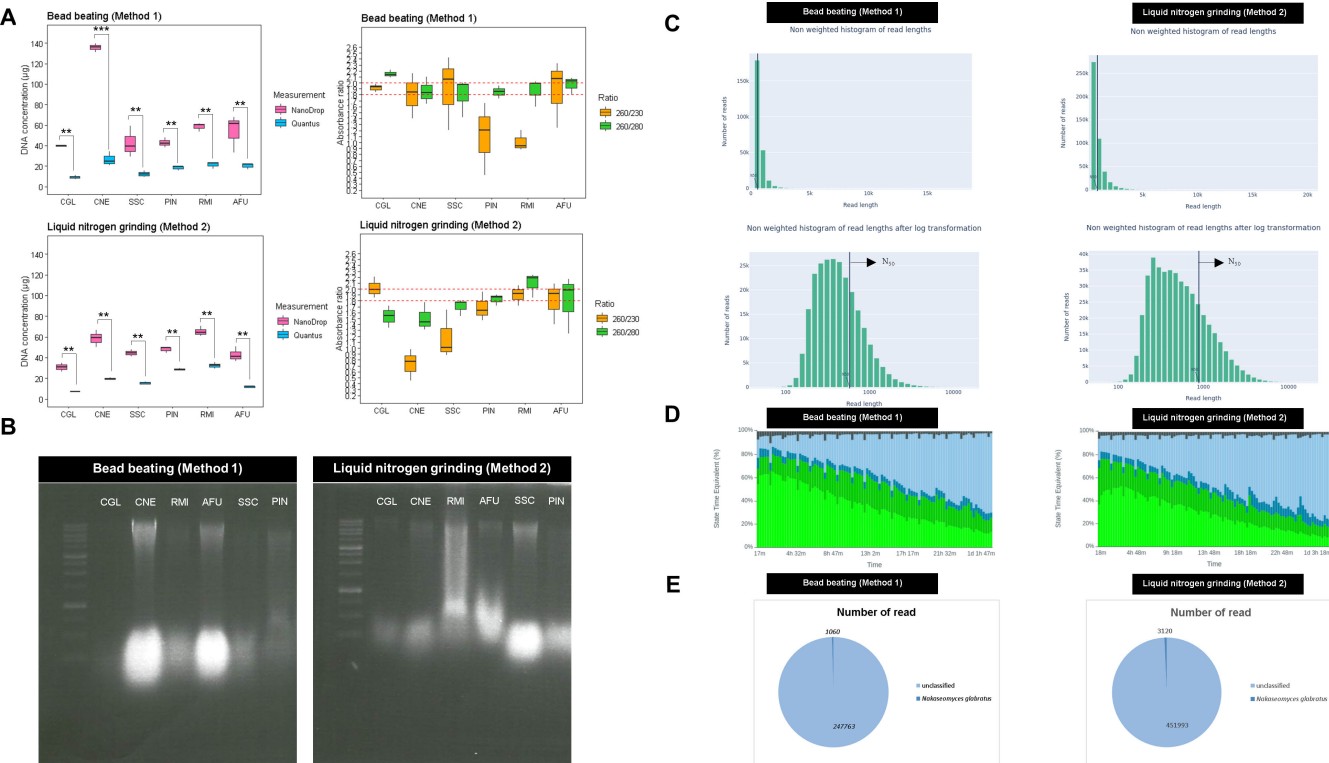

**FIG 1** Evaluation of the physical lysis efficacy for the pure cultures of six tested species including *Nakaseomyces glabratus* = CGL, *Cryptococcus neoformans* = CNE, *Aspergillus fumigatus* = AFU, *Sporothrix schenckii* = SSC, *Rhizopus microsporus* = RMI, and *Pythium insidiosum* = PIN in the figure, determined by (A) the quantity of the extracted DNA was determined using NanoDrop spectrophotometer and the Quantus fluorometer, with the purity assessed by NanoDrop, (B) DNA integrity was evaluated through 0.8% agarose gel electrophoresis, (C) Nanopore's pore status was checked using MinKNOW for DNA extracted with two different lysis methods: bead beating (Method 1) and liquid nitrogen grinding (Method 2), (D) read lengths generated from the two extraction methods were analyzed and visualized using Nanoplot, and (E) the identification results of *Nakaseomyces glabratus* were obtained through a physical lysis test. The level of significance was indicated by asterisks: *$P$-value < 0.05, **$P$-value < 0.01, and ***$P$-value < 0.001.

inactive pores (light blue in Fig. 1D). The proportion of active sequencing pores (dark and light green in Fig. 1D) was below 60% in both conditions, with inactivity increasing as time progressed. BLAST (59) identification results are presented in Fig. 1E.

## Chemical lysis using CTAB/SDS lysis generates relatively high molecular weight DNA from pure fungal culture

The amounts of DNA obtained in Method 3, as measured by the Quantus fluorometer, were 0.33 ± 0.02 µg for *N. glabratus*, 0.12 ± 0.01 µg for *C. neoformans*, 0.22 ± 0.04 µg for *S. schenckii*, 0.16 ± 0.03 µg for *P. insidiosum*, 0.24 ± 0.04 µg for *R. microsporus*, and 0.22 ± 0.01 µg for *A. fumigatus*, respectively. The amounts of DNA obtained with Method 4 were 1.53 ± 0.03 µg for *N. glabratus*, 0.15 ± 0.04 µg for *C. neoformans*, 0.46 ± 0.01 µg for *S. schenckii*, 0.23 ± 0.01 µg for *P. insidiosum*, 0.74 ± 0.02 µg for *R. microsporus*, and 0.62 ± 0.01 µg for *A. fumigatus* (Fig. 2A). Comparing Method 3 and Method 4 revealed a significant difference in DNA yield only for *N. glabratus* ($P$-value < 0.001, 95% CI, Bonferroni-corrected).

The CTAB lysis (Method 5) significantly increased DNA yield compared to Method 3 across all tested species ($P$-value < 0.001, 95% CI, Bonferroni-corrected) (Fig. 2A). The DNA amounts for Method 5 were 9.39 ± 0.43 µg for *N. glabratus*, 5.03 ± 0.08 µg for *C. neoformans*, 4.34 ± 0.56 µg for *S. schenckii*, 4.01 ± 0.75 µg for *P. insidiosum*, 4.67 ± 0.40 µg for *R. microsporus*, and 5.40 ± 0.56 µg for *A. fumigatus*.

The combined CTAB and SDS method (Method 6) further improved the lysis efficiency, yielding higher DNA amounts than Method 5 for all isolates ($P$-value < 0.001,

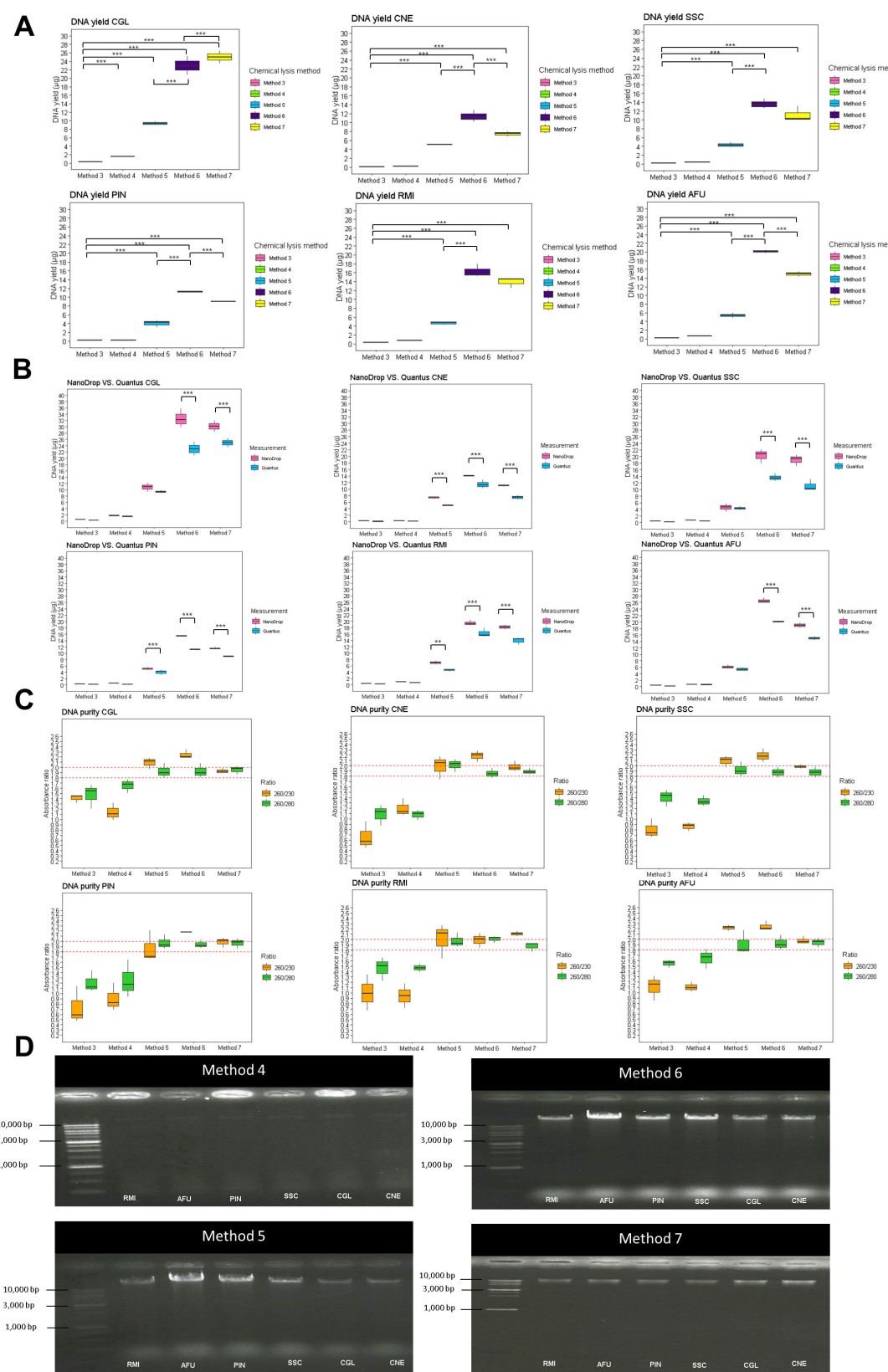

**FIG 2** Comparison of the efficacy of five chemical lysis buffers (Methods 3–7), each with different chemical components, using pure cultures of six tested species including *Nakaseomyces glabratus* = CGL, *Cryptococcus neoformans* = CNE, *Aspergillus fumigatus* = AFU, *Sporothrix schenckii* = SSC, *Rhizopus microsporus* (Continued on next page)

Fig 2 (Continued)

= RMI, and *Pythium insidiosum* = PIN, determined by (A) the amount of the DNA extracted from each method for each strain was determined using the Quantus fluorometer, (B) the comparison between DNA quantities measured with the NanoDrop spectrophotometer and the Quantus fluorometer, (C) the purity of the extracted DNA for each strain was assessed using the NanoDrop spectrophotometer, and (D) the integrity of the extracted DNA for each method (excluding Method 1) was evaluated through visualization with 0.8% agarose gel electrophoresis. The level of significance was indicated by asterisks: *$P < 0.05$, **$P < 0.01$, and ***$P < 0.001$.

95% CI, Bonferroni-corrected) (Fig. 2A). The DNA yields with Method 6, along with their 95% CIs, were 22.99 ± 2.22 µg for *N. glabratus*, 11.38 ± 1.29 µg for *C. neoformans*, 13.62 ± 1.12 µg for *S. schenckii*, 11.20 ± 0.170 µg *P. insidiosum*, 16.30 ± 1.47 µg for *R. microsporus*, and 20.12 ± 0.32 µg for *A. fumigatus*.

However, Method 7 (CTAB + SDS + Proteinase K + RNase A) showed a significant decrease in DNA yield for *C. neoformans*, *P. insidiosum*, and *A. fumigatus* (*P*-value < 0.001, 95% CI, Bonferroni-corrected) (Fig. 2A). The DNA yields with Method 7 were 24.99 ± 1.48 µg for *N. glabratus*, 7.44 ± 0.58 µg *C. neoformans*, 11.11 ± 1.72 µg for *S. schenckii*, 8.97 ± 0.05 µg for *P. insidiosum*, 13.88 ± 1.21 µg for *R. microsporus*, and 14.96 ± 0.56 µg for *A. fumigatus*.

DNA quantification using NanoDrop spectrophotometer and Quantus fluorometer revealed significant discrepancies (Fig. 2B). The result from Method 5 (CTAB + Proteinase K), *R. microsporus* showed a mean of 7.01 ± 0.46 µg (NanoDrop spectrophotometer) compared to 4.67 ± 0.40 µg (Quantus fluorometer) (*P*-value < 0.01, 95% CI, Bonferroni-corrected). Similarly, *P. insidiosum* yielded 5.15 ± 0.42 µg vs 4.00 ± 0.75 µg, and *C. neoformans* yielded 7.35 ± 0.32 µg vs 5.03 ± 0.08 µg (*P*-value < 0.001, 95% CI, Bonferroni-corrected).

Using Method 6 (CTAB + SDS), the DNA yielded 32.62 ± 3.06 µg for *N. glabratus*, 14.04 ± 0.10 µg for *C. neoformans*, 20.24 ± 2.14 µg for *S. schenckii*, 15.42 ± 0.12 µg for *P. insidiosum*, 19.43 ± 0.84 µg for *R. microsporus*, and 26.56 ± 0.77 µg for *A. fumigatus* measured with NanoDrop spectrophotometer. These values were significantly higher than those measured with Quantus fluorometer (*P*-value < 0.001, 95% CI, Bonferroni-corrected). Method 7 showed similar discrepancies as Method 6 for the DNA concentrations obtained (*P*-value < 0.001, 95% CI, Bonferroni-corrected).

The DNA purity was evaluated using the 260/280 and 260/230 ratios (Fig. 2C). The results from Methods 1 and 2 were suboptimal, as evidenced by lower absorbance ratios. In contrast, DNA obtained from Methods 5–7 showed acceptable purity with 260/280 ratios ranging from 1.8 to 2.0 and 260/230 ratios exceeding 2.0.

The DNA integrity was better preserved in Methods 5–7, with SDS addition increasing the amount of high molecular weight DNA (Fig. 2D). The DNA from SDS/CTAB method (Method 6) had greater lysis capacity for both hyphal and yeast cells (Fig. 3B) compared to those treated with SDS + Proteinase K (Method 4) (Fig. 3A). No DNA shearing was detected, as intact DNA bands were visible on 0.8% agarose gel electrophoresis (Table S1). Based on its superior yield, quality, and integrity, Method 6 was identified as the optimal lysis method.

## DNA extraction using mobilized paramagnetic beads was optimal for DNA extraction from both pure fungal culture and simulated clinical specimens with the application of a human DNA depletion protocol

The aim of our study was to decrease contamination, leading to the better purity of extracted DNA. This study employed both fixed columns (spin column technique) and mobile magnetic particles as a solid phase, to isolate DNA from pure cultures of *N. glabratus* representing yeasts and *A. fumigatus*, representing molds.

In our study, *A. fumigatus* exhibited a significantly lower DNA yield (*P*-value < 0.05, 95% CI, Bonferroni-corrected) when compared to that obtained from the magnetic beads (15.34 ± 0.23 µg) and from a spin column (15.46 ± 1.39 µg), as opposed to 18.45 ± 1.11 µg obtained with Method 6. However, it is important to note that both methods proved equally effective in purifying DNA from both isolates (Fig. 4A). In the case of *N. glabratus*,

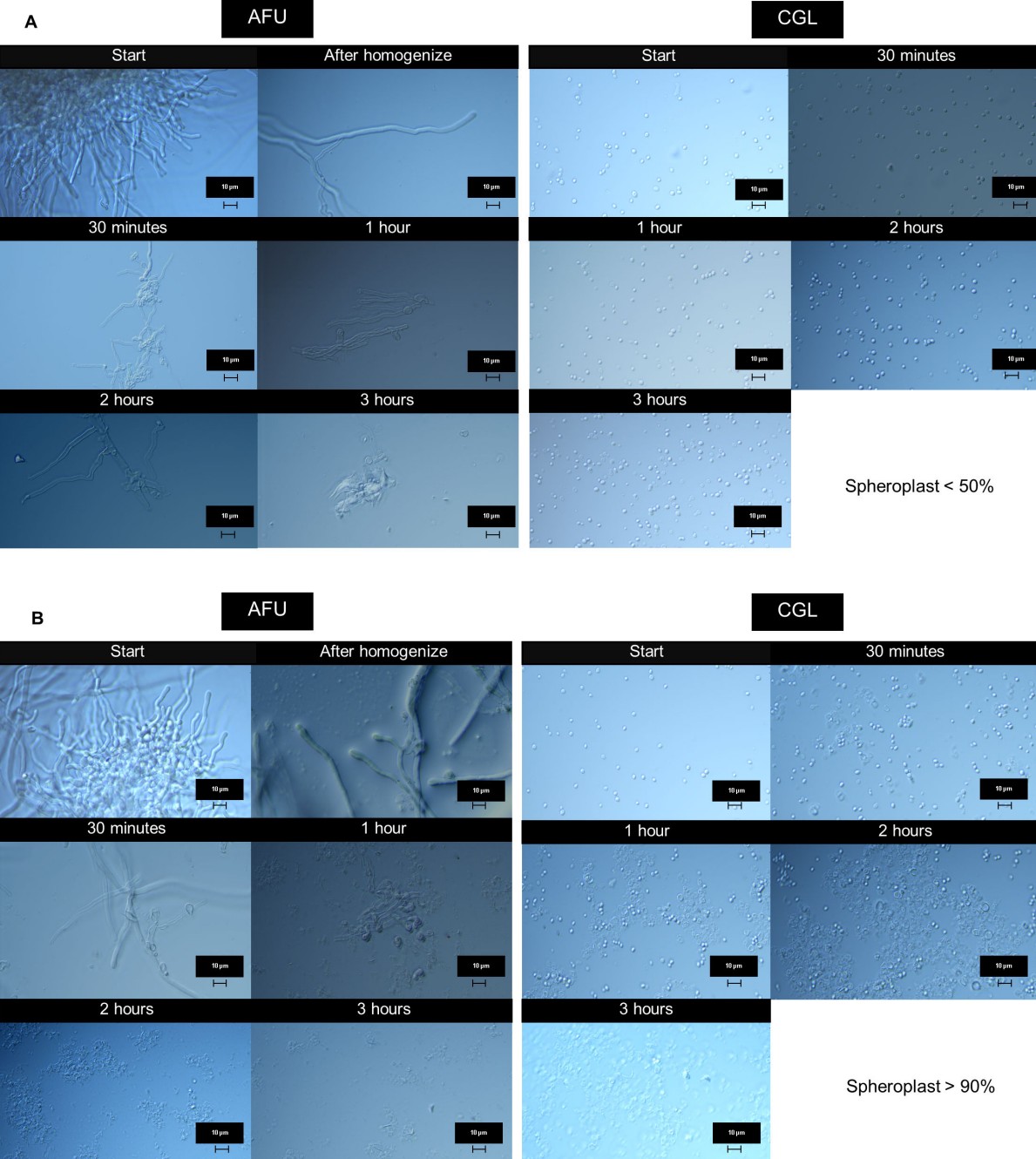

**FIG 3** Microscopic examination of the lysate was conducted on *Nakaseomyces glabratus* = CGL and *Aspergillus fumigatus* = AFU as representative models of yeast and mold to compare the degree of lysis between (A) Method 4 (SDS with Proteinase K) and (B) Method 6 (SDS/CTAB with Proteinase K) using differential interference contrast microscopy. Structural changes in the fungal cells were observed at 30 minutes, 1 hour, 2 hours, and 3 hours after treatment with the lysis buffer for both the filamentous mold and yeast.

comparable yields (17.96 ± 0.89 µg DNA for magnetic beads and 19.72 ± 0.49 µg for spin column compared with 20.93 ± 1.70 µg of Method 6) were also observed (Fig. 4A). The yield of intact DNA (Fig. 4B) demonstrated the efficiency of both methods. Furthermore, both methods yielded acceptable purities, as indicated by the 260/280 and 260/230 ratios (Fig. 4C). These results suggest that the silica-based chromatography methods are effective for purifying DNA from pure cultures of the tested isolates and could be a valuable addition to various downstream applications.

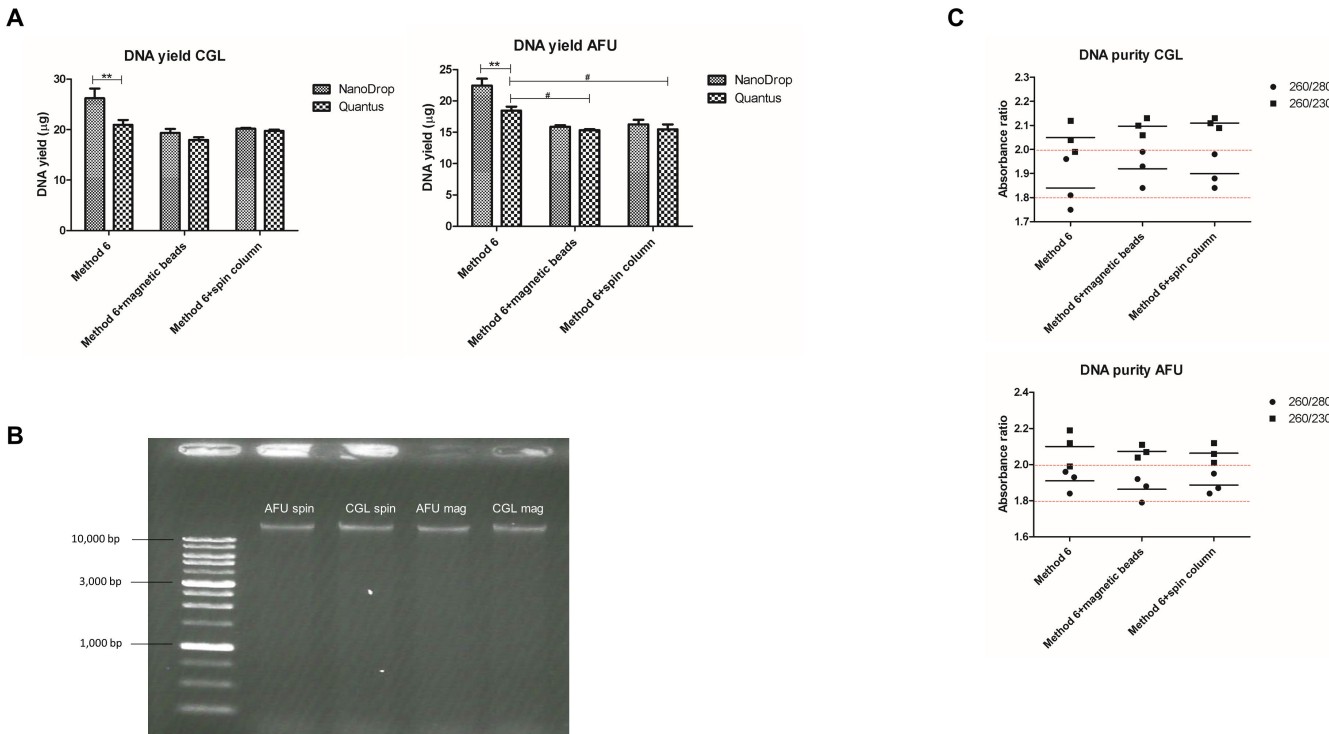

**FIG 4** Comparison of two silica-based chromatography clean-up protocols—the paramagnetic beads method (magnetic beads) and the spin-column method (spin column)—using the same lysis method (Method 6) without clean-up steps. The comparison included (A) the yield of extracted DNA measured with NanoDrop spectrophotometer and Quantus fluorometer; (B) the integrity of the extracted DNA, and (C) the purity of the extracted DNA for each representative mold and yeast strain. (*Aspergillus fumigatus* = AFU and *Nakaseomyces glabratus* = CGL, respectively) included in this study. The level of significance within the same group was indicated by asterisks: *$P$-value < 0.05, **$P$-value < 0.01, and ***$P$-value < 0.001 and the level of significance between groups was indicated by hashtags: #$P$-value < 0.05, ##$P$-value < 0.01, and ###$P$-value < 0.001.

Moreover, *N. glabratus* spiked at the concentration of $10^7$ and $10^8$ cells/mL in 1 mL EDTA blood and applying the human DNA depletion protocol was evaluated. At a concentration of $10^7$ cells/mL, the DNA yielded from the spin-column technique was significantly less than that obtained from the paramagnetic beads, based on the NanoDrop measurements (730 ± 33.87 ng for the spin-column technique vs 1,071.33 ± 179.70 ng for the paramagnetic beads technique) ($P$-value < 0.01, 95% CI, Bonferroni-corrected) and using Quantus (774.63 ± 45.87 ng for the spin-column technique vs 1,014.33 ± 122.77 ng for the paramagnetic beads technique) ($P$-value < 0.05, 95% CI, Bonferroni-corrected). Similar observations were found for the spiked concentration of $10^8$ cells/mL. Comparing the DNA concentrations, based on NanoDrop spectrophotometer measurements, obtained from the spin-column technique (1,184.67 ± 93.93 ng) with that obtained from the paramagnetic bead technique (1,794.67 ± 1,74.35 ng) ($P$-value < 0.001, 95% CI, Bonferroni-corrected). The DNA concentration from the spin-column approach was lower, measured by Quantus (1,117.87 ± 29.05 ng vs 1,743.67 ± 29.05 ng of spin column and paramagnetic beads, respectively) ($P$-value < 0.001, 95% CI, Bonferroni-corrected) as shown in Fig. 5A and B. Moreover, intact DNA was undetectable after 0.8% agarose gel electrophoresis for the $10^7$ cells/mL spiked yeast sample, suggesting that shearing of the DNA occurs when using lower than $10^8$ cells/mL for the DNA extraction (Fig. 5C).

## Efficacy of the human DNA depletion protocol

The simulated hemoculture was tested with and without spiking with *N. glabratus* and with and without applying human DNA depletion. Although DNA amounts were not

significantly different between the groups when measured by NanoDrop spectrophotometer and Quantus fluorometer (Fig. 6A), bands on 0.8% agarose gel electrophoresis revealed a lack of intact DNA in hemoculture treated with the human DNA depletion protocol compared to untreated hemoculture (Fig. 6B).

Interestingly, the read numbers from the hemoculture alone treated with the human DNA depletion protocol were significantly higher (1,226,023 ± 125,014 reads) than in the untreated (86,751 ± 26,778 reads) condition before any length filtering (*P*-value < 0.01, 95% CI, Bonferroni-corrected) (Fig. 6C). However, after applying a read length filter with Filtlong ver. 0.2.1 (https://github.com/rrwick/Filtlong), the number of reads shorter than 1,000 bp was higher in the treated hemoculture (386,782 ± 78,302 reads) compared to the untreated one (32,343 ± 818 reads) (Fig. 6C). In addition, the number of longer reads (10,000 bp) in the treated condition (27 ± 7 reads) was drastically reduced compared to the untreated condition (7,153 ± 516 reads) (Fig. 6C). BLAST (59) classification showed that the number of reads identified as human was reduced with the depletion protocol, especially for reads longer than 10,000 bp (Fig. 6D).

In the yeast-spiked hemoculture, the application of the human DNA depletion protocol yielded some notable results. After human DNA depletion, fungal DNA comprised 99.53% of the total reads, with 174,476 ± 130,656 reads classified as *N. glabratus*, compared to 813 ± 572 reads in the untreated condition (*P*-value < 0.05, 95% CI, Bonferroni-corrected) (Fig. 6E). Meanwhile, human reads decreased by 88.73% in the treated condition, with 174 ± 48 reads classified as human, compared to 1,544 ± 40 reads in the untreated condition (*P*-value < 0.05, 95% CI, Bonferroni-corrected) (Fig. 6E). Additionally, the $N_{50}$ value for read length in hemoculture alone treated with the human DNA depletion protocol (2,061 ± 604 bp) was significantly lower than that in the untreated condition (13,151 ± 1,207 bp), (*P*-value < 0.01, 95% CI, Bonferroni-corrected).

**A**

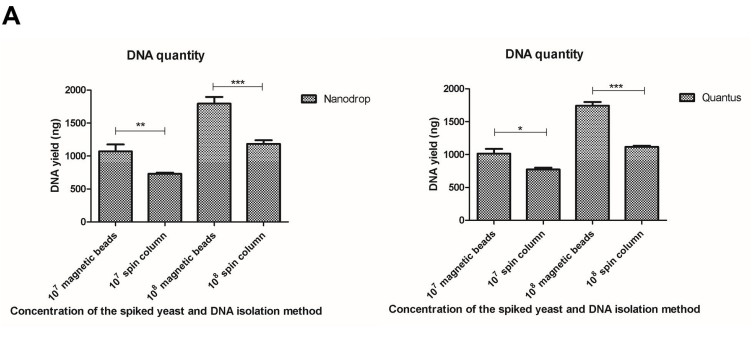

**B**

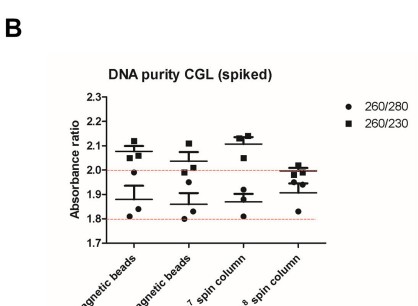

**C**

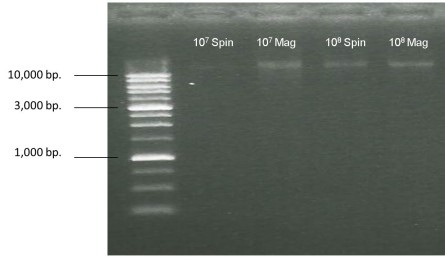

**FIG 5** Comparison of two different silica-based chromatography DNA extraction and clean-up protocols, the paramagnetic beads method (magnetic beads = Mag) and the spin-column method (spin column = Spin) in connection with the same lysis method (Method 6) for blood spiked with the yeast (*Nakaseomyces glabratus*), showing in (A) the yield of the extracted DNA measured with NanoDrop and Qauntus, (B) the integrity of the extracted DNA, and (C) the purity of the extracted DNA. The level of significance was indicated by asterisks: *P*-value < 0.05, **P*-value < 0.01, and ***P*-value < 0.001.

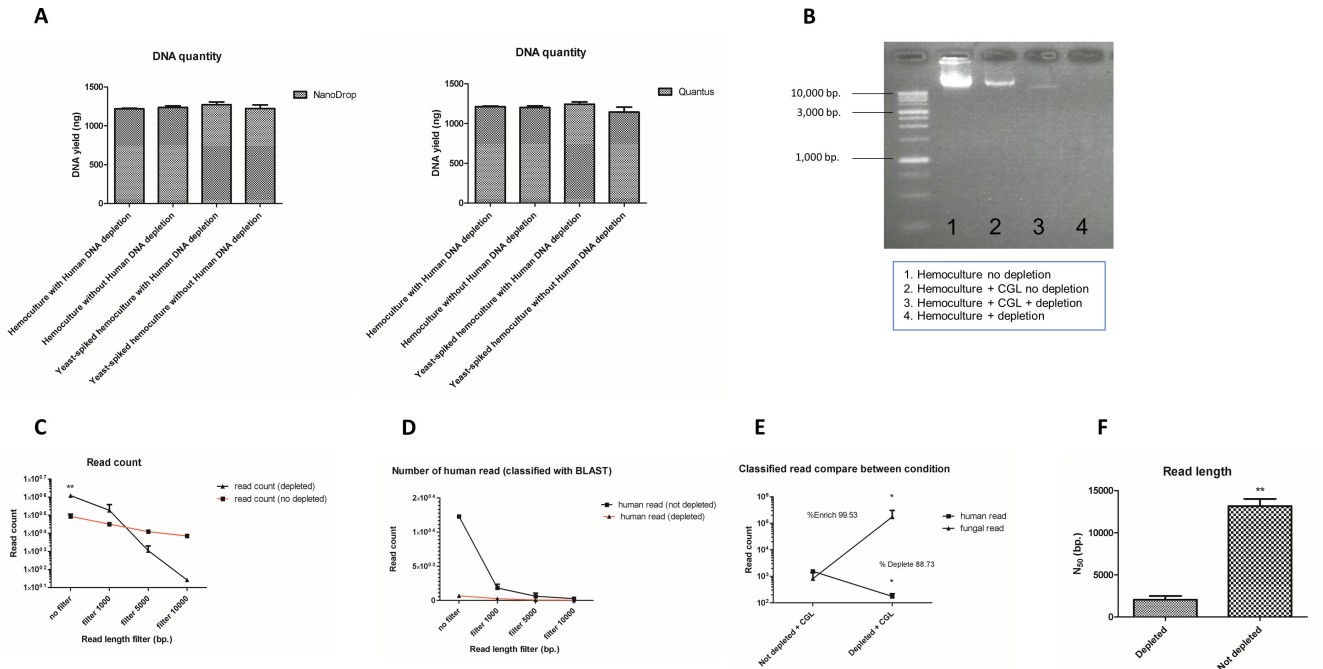

**FIG 6** Human DNA depletion efficacy was tested in simulated hemocultures with and without the addition of yeast (*Nakaseomyces glabratus* = CGL) at a concentration of 200 cells/mL before incubation: (A) DNA quality and quantity in both conditions of simulated hemoculture, with and without the application of the human DNA depletion protocol; (B) the presence of intact DNA in the extracted samples from both conditions of simulated hemoculture, with and without the application of the human DNA depletion protocol; (C) the number of read counts in the extracted DNA from simulated hemoculture (no spiked yeast), with and without the application of the human DNA depletion protocol, under read length filtering conditions of 1,000 bp, 5,000 bp, 10,000 bp, and no filtering; (D) the number of human reads classified by BLAST in the extracted DNA from simulated hemoculture (no spiked yeast), with and without the application of the human DNA depletion protocol, under read length filtering conditions of 1,000 bp, 5,000 bp, 10,000 bp, and no filtering; (E) the number of fungal and human reads in simulated hemoculture (with spiked yeast) with and without the application of the human DNA depletion protocol; and (F) a comparison of read lengths in simulated hemoculture (no spiked yeast) with and without the human DNA depletion protocol. The level of significance was indicated by asterisk: *$P$-value < 0.05, **$P$-value < 0.01, and ***$P$-value < 0.001.

## Efficacy of the optimized DNA extraction protocol for sequencing with long-read ONT when applied to simulated clinical specimens

Based on the above results, a human DNA depletion protocol in connection with the herein developed extraction protocol (Method 6) using the SDS/CTAB lysis buffer followed by paramagnetic bead clean-up was tested with simulated hemoculture specimens of all six fungal species. The result showed that when DNA was extracted with the optimized method and visualized on 0.8% agarose gel electrophoresis, all simulated hemocultures displayed the presence of intact DNA, as shown in Fig. 7A. When this DNA was sequenced on the MinION, 80% of the pores were active (light green pores in Fig. 7B) and less than 10% were inactive (light blue pores in Fig. 7B), indicating a high sequencing run efficacy and a good quality of the prepared library. Furthermore, the high molecular weight DNA with an average $N_{50}$ across all six species resulted in about 10.137 kb (4.261 kb for *R. microsporus*, 11.915 kb for *A. fumigatus*, 9.192 kb for *P. insidiosum*, 15.785 kb for *S. schenckii*, 7.103 kb for *N. glabratus*, and 4.238 kb for *C. neoformans*) (Fig. 7C). The over 10 kb fragments yielded 62 reads for *R. microsporus*, 2,145 reads for *A. fumigatus*, 6,385 reads for *P. insidiosum*, 1,408 reads for *S. schenckii*, 2,016 reads for *N. glabratus*, and 2,089 reads for *C. neoformans*.

It was found that fungal species-specific reads were classified more than human reads in all species, as shown in Fig. 7D. The assembly statistics analysis of the fragments was summarized in Table 2. The $N_{50}$ of fragments were 15,928 kb for *R. microsporus*, 48,846 kb for *A. fumigatus*, 35,436 kb for *P. insidiosum*, 34,843 kb for *S. schenckii*, 87,110 kb for *N.*

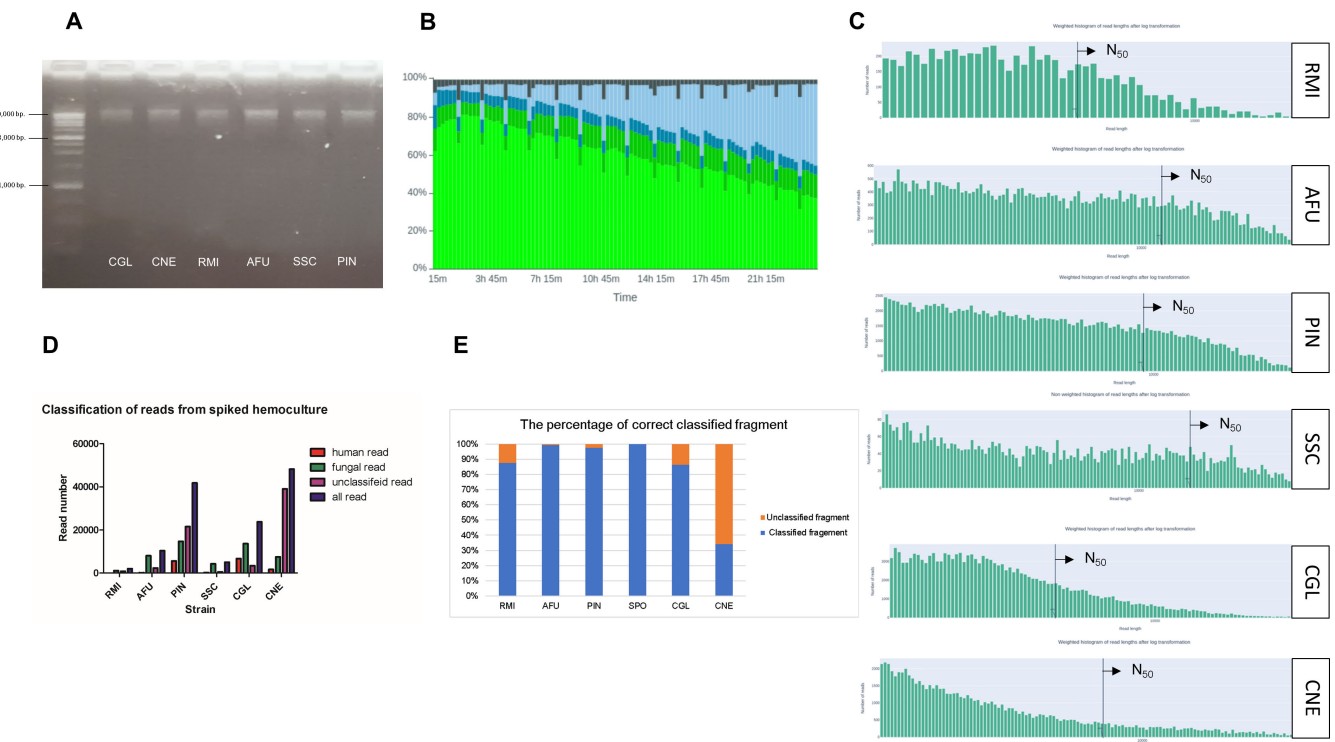

**FIG 7** Evaluating the optimized DNA extraction and pre-processing protocol using simulated hemocultures spiked with six species including *Nakaseomyces glabratus* = CGL, *Cryptococcus neoformans* = CNE, *Aspergillus fumigatus* = AFU, *Sporothrix schenckii* = SSC, *Rhizopus microsporus* = RMI, and *Pythium insidiosum* = PIN: (A) the presence of intact DNA in the extracted samples from all six simulated hemocultures, (B) Checking the pore status using MinKNOW for the DNA processed and extracted with the optimized protocol, (C) the read length distribution of DNA extracted from all six simulated hemocultures using the optimized protocol, (D) the classification of reads from all six simulated hemocultures extracted using the optimized protocol, filtered for read lengths greater than 1,000 bp, and (E) the percentage of correctly classified fragments for each strain after assembly.

*glabratus*, and 10,105 kb for *C. neoforman*s. The largest fragments were 35,571 kb for *R. microsporus*, 171,033 kb for *A. fumigatus*, 249,410 kb for *P. insidiosum*, 155,266 kb for *S. schenckii*, 483,595 kb for *N. glabratus*, and 56,465 kb for *C. neoformans*. The percentage of correctly classified sequence fragments after assembly compared with all sequence fragments generated after assembly were 87.5% (*R. microsporus*), 99.20% (*A. fumigatus*), 97.58% (*P. insidiosum*), 100.00% (*S. schenckii*), 86.26% (*N. glabratus*), and 34.12% (*C. neoformans*), respectively (Fig. 7E).

## DISCUSSION

Extracting DNA from fungi is notoriously challenging, often leading to suboptimal yield, purity, and integrity. For the ONT, the optimal quality, with a NanoDrop 260/280 ratio of 1.8–2.0 and a 260/230 ratio of 2.0–2.2 is recommended. However, due to the diverse composition of the fungal cell walls, enzymatic lysis methods like zymolyase or lyticase

**TABLE 2** Summary of the assembly statistics for simulated hemocultures of each fungal species in this study

| Species | $N_{50}$ | Largest fragment (base pair) |
| --- | --- | --- |
| *Rhizopus microsporus* | 15,928 | 35,571 |
| *Aspergillus fumigatus* | 48,846 | 171,033 |
| *Pythium insidiosum* | 35,436 | 249,410 |
| *Sporothrix schenckii* | 34,843 | 155,266 |
| *Nakaseomyces glabratus* | 87,110 | 483,595 |
| *Cryptococcus neoformans* | 10,105 | 56,465 |

often produce insufficient yield and purity (20, 62, 63). Additionally, enzymatic methods risk contaminating fungal DNA (27, 64). To address this, physical methods like bead beating and liquid nitrogen grinding can be used to effectively disrupt fungal cell walls as they tend to work well across various groups of fungi (24, 65). Our study found that liquid nitrogen grinding and bead beating provided the highest DNA yield with the best quality and quantity, as shown by NanoDrop spectrophotometer and Quantus fluorometer measurement (Fig. 1A). However, these methods also resulted in significant contamination and potential DNA damage due to shear forces (24, 66) as shown in Fig. 1B and C. These methods are especially effective for extracting fungal DNA for targeted sequencing, such as the ITS, which do not need intact DNA (34, 65, 67). Some studies show that using liquid nitrogen grinding can reduce heat and shear force, preserving intact DNA (68–70). However, using this method in clinical labs can be challenging due to its hazards and the need for meticulous preparation and cleaning. Sterile tools are required for each sample, leading to longer processing times and preventing batch analysis which is not user-friendly. Our findings also highlight that liquid nitrogen grinding requires careful optimization to avoid DNA shearing (Fig. 1B).

When using physical lysis methods like bead beating or liquid nitrogen grinding, the lysate may contain various contaminants, as these methods do not selectively remove proteins, polysaccharides, or pigments. Although absorbance ratios can initially reveal the contaminants, further purification steps are needed. Phenol/chloroform extraction separates compounds based on polarity, working well with mammalian cells by denaturing proteins and lipids (71). However, during the DNA extraction from fungal or plant cells, polysaccharides and phenolic compounds can co-precipitate with nucleic acids (72), affecting sequencing by increasing inactive nanopores (73). Our study showed that combining physical lysis with phenol/chloroform extraction resulted in lower DNA purity, leading to more inactive pores during sequencing, with the active pore count dropping from 60% to less than 20% within 24 hours due to potential residue from the extraction process.

Chemical lysis preserves DNA integrity by using chemicals to break down cell components. For mammalian cells, detergents like SDS or Triton X-100 disrupt the phospholipid bilayer, while SDS also denatures proteins, aiding Proteinase K in protein digestion (74). Additional components, like NaCl, Tris-HCl, and EDTA, optimize ionic strength, pH, and inhibit DNases (75). We used these chemical lysis methods and found that they were not quite effective for fungal lysis, as shown in Fig. 2A.

To extract high molecular weight DNA from fungi, additional components in the lysis buffer are needed, particularly to address the high number of polysaccharides in the cell wall structure. The common practice for chemical lysis, e.g., being used for DNA extraction from plants, often involves using CTAB-PVP (32–34, 76). CTAB, a cationic detergent, precipitates nucleotides and polysaccharides at lower salt concentrations (usually <0.4 M) (30, 77) but may co-precipitate other macromolecules, affecting DNA purity (78, 79). Using CTAB with higher salt concentrations or combining it with SDS enhances DNA yield by precipitating polysaccharides while keeping DNA in the supernatant as shown for some other polysaccharide-rich components (32–34, 69, 80) and was also shown in our study for fungal cells or hyphae (Fig. 2A). Adding PVP improves the yield by removing phenolic compounds that degrade DNA (81–84). Ultracentrifugation helps remove contaminants, leaving mostly nucleic acids in the aqueous phase (85). Our study found that the CTAB/SDS lysis buffer treatment significantly increased DNA yield and purity compared to other methods. Despite this, fungal hyphae remain difficult to lyse chemically.

The lysis method optimized in our study significantly improves DNA yield and integrity compared to previous fungal DNA extraction methods, which often rely on CTAB-PVP lysis with liquid nitrogen grinding or lyophilization (8, 19, 24, 54, 84). The previous methods have proven to be impractical for routine medical labs due to contamination risks and logistical challenges. The herein optimized method, which incorporates SDS, has shown superior results in extracting fungal DNA and is safer by

reducing the need for hazardous agents like liquid nitrogen. Our study shows that the SDS-CTAB approach yields more fungal DNA than some previous studies (34, 69, 82, 86–88), using either chemical lysis strategies, including SDS, CTAB, or even SDS-CTAB with or without liquid nitrogen grinding, of which some were also applied for DNA extraction as basis for NGS (82, 86).

Moreover, we observed that using RNase A (Method 5) led to a decrease in DNA yield, possibly due to DNA-RNase A binding, which complicates DNA recovery (89, 90). Given these factors and the difficulty of inactivating and removing the highly stable RNase A enzyme from the system despite the application of high temperatures, various pH ranges, or the presence of EDTA (91), RNase A treatment is not suitable for high molecular weight DNA isolation from fungi. Therefore, the SDS/CTAB method (Method 6) herein developed is optimal for cell lysis. Although ultracentrifugation in Method 6 or 7 yielded high amounts of DNA, the presence of co-precipitated contaminants was evident, suggesting technical or procedural challenges in removing impurities.

Silica-based chromatography, using either paramagnetic beads or spin columns, effectively yields intact DNA from pure fungal cultures (Fig. 4A through C). However, in yeast-spiked blood samples, the use of spin columns resulted in significantly lower DNA recovery and integrity compared to the use of paramagnetic beads (Fig. 5A), especially at lower yeast concentrations. There has been a study showing that the amount of the microbe DNA recovery via the spin column method was lower by almost three times compared to the magnetic bead procedure (92). Another study supports our findings that magnetic beads not only yield more DNA than the spin column but also preserve the integrity of the DNA better than the spin column (93). This reduction could be due to the shear force from centrifugation and potential occlusion in spin columns, which are less compatible with clinical specimens (94). Paramagnetic beads offer better DNA recovery and preservation, making them more suitable for fungal DNA extraction from clinical samples, and thus being the preferred method.

A major challenge in extracting fungal DNA from clinical samples is the contamination of host DNA, which can interfere with pathogen detection. Our protocol, modified from Charalampous et al. (40), was designed to deplete the host DNA. This protocol had already been applied in a wide variety of clinical samples including respiratory samples (BAL, sputum, ETA, and blood) (40, 41, 46, 95, 96) and has already proven to work with a wide range of microbes including bacteria, fungi, and mycobacteria (41, 46). However, for fungal hemocultures, the implementation of the protocol is still lacking (17, 46). The protocol uses saponin to selectively lyse mammalian cells by disrupting their membranes and causing osmotic shock. HL-SAN DNase is effective in high salt conditions but is inhibited by heat and cleaves DNA into shorter fragments (40).

To monitor potential contaminants in the DNA to be used for metagenomics, control conditions were established, including hemoculture alone, hemoculture with spiked yeast (both with and without the human DNA depletion protocol), and additional tests involving other fungi spiked into hemoculture. Our study confirmed that this protocol is an effective combination with the herein-used lysis and DNA isolation methods, retaining high DNA yield, purity, and integrity (Fig. 5A through C). Read-level analysis revealed that while human DNA was fragmented rather than eliminated, these obtained shorter fragments were less likely to be classified by BLAST (59), reducing interference with fungal DNA classification (Fig. 6C and D). The ratio of human DNA to fungal DNA, as well as the other spiked-in microbial DNA, was found that the residual human DNA contamination after human DNA depletion remained below 10% of the total reads (Fig. 7D). This result aligns with the levels of human DNA observed after treatment with the depletion protocol, as verified in our tests (Fig. 6D and E), supporting the protocol's effectiveness in minimizing human DNA interference. In yeast-spiked hemocultures, the human DNA depletion protocol improved fungal read classification, echoing findings from studies on bacteria and mycobacteria (42, 46). Despite some reads remaining unclassified, after assembly, likely due to limitations in the classifier, this protocol successfully enriched fungal DNA for more accurate pathogen detection (97).

Essentially, this study acknowledges a limitation in the detection of non-fungal DNA, which could potentially introduce other sources of contamination. However, identifying such DNA may be beyond the current study's scope as this study was aimed to initiate the development of a DNA extraction protocol from sterile site specimens for fungal infection. Moreover, despite the effort to cover the different fungal morphotypes, we acknowledge that the other limitation in this study is the limited strain of fungi included. To implement the protocol in the service routine clinical laboratory, larger cohort should be done in the future. To lay the initial work for developing direct pathogenic fungi identification from clinical samples, this study could potentially provide the fundamental pipeline for DNA preparation for detecting fungi via metagenomics, including simultaneous detection of antifungal drug response profiles, through molecular techniques using ONT in the future. However, as for the advancements in fungal diagnostics, developing an appropriate bioinformatic pipeline for antifungal drug responses will be essential. This includes building a database of genetic markers that correlate with different antifungal response patterns. Moreover, it will be necessary to explore targeted extraction of specific genomic regions with adequate coverage to capture accurate genetic profiles as part of future untargeted shotgun metagenomic testing efforts.

## Conclusion

An optimized streamlined DNA extraction protocol (Method 6) for fungal DNA from clinical samples, using SDS/CTAB lysis, silica-based paramagnetic beads, and human DNA depletion as outlined in Fig. 8 has herein been developed. When tested on simulated hemocultures, this method produced high molecular weight DNA with an $N_{50}$ over 10 kb, compatible with ONT sequencing. The protocol is efficient, eliminating hazardous steps like liquid nitrogen grinding and phenol-chloroform extraction, and is automated with minimal hands-on time, reducing the extraction process to under 6 hours (Table S2). This time may be reduced further with the development of adaptive sampling to

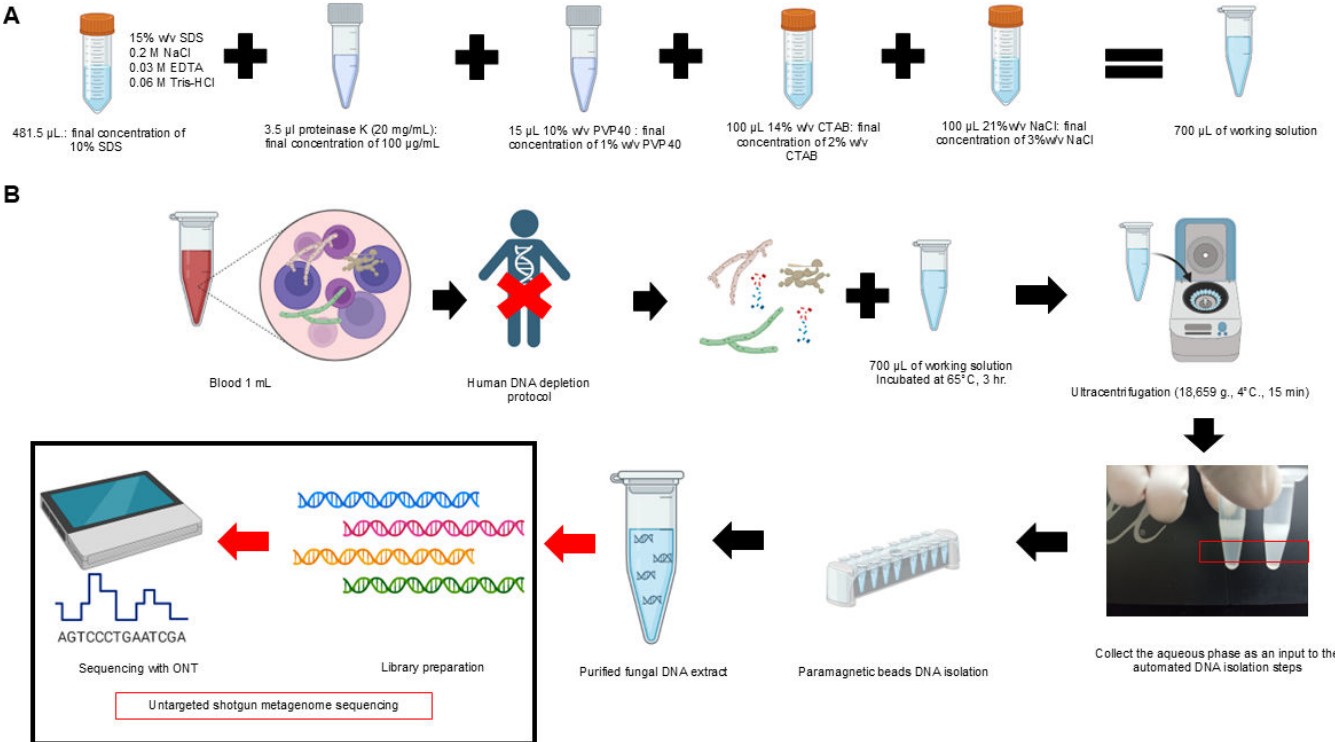

**FIG 8** Proposed DNA extraction and pre-processing process: (A) the proposed lysis buffer component (Method 6), and (B) the full DNA extraction and pre-processing pipeline for DNA extraction of fungi from clinical specimen used for untargeted shotgun metagenome sequencing. The illustration was generated via https://www.biorender.com.

decrease the human DNA bioinformatically. While suitable for long-read sequencing, the protocol does not generate ultra-long DNA for whole-genome assembly. Further validation on different specimen types is recommended.

In conclusion, this protocol offers a practical and efficient alternative for fungal DNA extraction in clinical settings, facilitating long-read sequencing for fungal identification. As ONT technology advances, this method could lead to even more automated, bioinformatics-driven workflows in clinical diagnostics.

## ACKNOWLEDGMENTS

We thank the Department of Microbiology, Faculty of Medicine, Chulalongkorn University, for providing the laboratory and computational facilities used in our research. We also thank the Chulalongkorn Medical Research Center (Chula MRC), Faculty of Medicine, Chulalongkorn University, for providing the equipment used in this study.

This work was supported by the National Research Council of Thailand (N11A650143 to A.C. and W.M.), the Faculty of Medicine, Chulalongkorn University (RA-MF 47/64 to A.C.), and Thailand Science Research and Innovation Fund Chulalongkorn University (HEA663000027).

## AUTHOR AFFILIATIONS

[1]Department of Microbiology, Faculty of Medicine, Chulalongkorn University, Bangkok, Thailand

[2]Westerdijk Fungal Biodiversity Institute, Utrecht, the Netherlands

[3]Molecular Mycology Research Laboratory, Centre for Infectious Diseases and Microbiology, Westmead Clinical School, Sydney Medical School, Faculty of Medicine and Health, Sydney Infectious Diseases Institute, University of Sydney, Westmead Hospital, Research and Education Network, Westmead, New South Wales, Australia

[4]Department of Transfusion Medicine and Clinical Microbiology, Faculty of Allied Health Sciences, Chulalongkorn University, Bangkok, Thailand

[5]Research Unit of Medical Mycology Diagnosis, Chulalongkorn University, Bangkok, Thailand

[6]Department of Microbiology, Faculty of Science, Chulalongkorn University, Bangkok, Thailand

[7]Department of Microbiology and Plant Pathology, University of California, Riverside, California, USA

[8]Department of Biomedical Informatics, College of Medicine, University of Arkansas for Medical Sciences, Little Rock, Arkansas, USA

[9]Division of Medical Bioinformatics, Faculty of Medicine, Siriraj Hospital, Mahidol University, Bangkok, Thailand

[10]National Center for Genetic Engineering and Biotechnology (BIOTEC), National Science and Technology Development Agency (NSTDA), Pathum Thani, Thailand

[11]Center of Excellence in Antimicrobial Resistance and Stewardship, Chulalongkorn University, Bangkok, Thailand

## AUTHOR ORCIDs

Nuttapon Pombubpa http://orcid.org/0000-0003-3385-5331
Thidathip Wongsurawat http://orcid.org/0000-0002-3659-2074
Piroon Jenjaroenpun http://orcid.org/0000-0002-1555-401X
Ariya Chindamporn http://orcid.org/0000-0002-9761-1670

## FUNDING

| Funder | Grant(s) | Author(s) |
| --- | --- | --- |
| National Research Council of Thailand | N11A650143 | Ariya Chindamporn |

| Funder | Grant(s) | Author(s) |
|---|---|---|
| Faculty of Medicine, Chulalongkorn University | RA-MF 47/64 | Ariya Chindamporn |
| Chulalongkorn University | HEA663000027 | Ariya Chindamporn |

## AUTHOR CONTRIBUTIONS

Nattapong Langsiri, Conceptualization, Formal analysis, Investigation, Methodology, Software, Visualization, Writing – original draft, Writing – review and editing | Wieland Meyer, Conceptualization, Funding acquisition, Methodology, Resources, Validation, Writing – review and editing | Laszlo Irinyi, Conceptualization, Data curation, Formal analysis, Funding acquisition, Investigation, Methodology, Resources, Software, Validation, Writing – review and editing | Navaporn Worasilchai, Conceptualization, Formal analysis, Funding acquisition, Investigation, Methodology, Project administration, Resources, Software, Supervision, Visualization, Writing – original draft, Writing – review and editing | Nuttapon Pombubpa, Data curation, Formal analysis, Investigation, Methodology, Software, Validation, Writing – review and editing | Thidathip Wongsurawat, Conceptualization, Formal analysis, Funding acquisition, Investigation, Methodology, Project administration, Resources, Supervision, Writing – original draft, Writing – review and editing | Piroon Jenjaroenpun, Data curation, Methodology, Resources, Software, Validation, Writing – review and editing | J. Jennifer Luangsa-ard, Formal analysis, Funding acquisition, Investigation, Methodology, Resources, Supervision, Writing – review and editing | Ariya Chindamporn, Conceptualization, Data curation, Funding acquisition, Methodology, Project administration, Resources, Software, Supervision, Validation, Writing – original draft, Writing – review and editing

## DATA AVAILABILITY

All the sequences have been deposited in the Sequence Read Archive under the BioProject ID PRJNA1077924 (simulated hemoculture of six species of fungal species), PRJNA1078070 (simulated hemoculture spiked with *N. glabratus* for testing human DNA depletion protocol and *N. glabratus* read using for testing the physical lysis method), and PRJNA1078071 (simulated hemoculture for testing human DNA depletion protocol).

## ADDITIONAL FILES

The following material is available online.

### Supplemental Material

**Table S1 (mSystems01166-24-s0001.docx).** The metadata of the six studied fungal isolates.
**Table S2 (mSystems01166-24-s0002.docx).** Comparison of the extraction steps of the newly designed DNA extraction protocol compare with the conventional DNA extraction protocol for fungi.

### Open Peer Review

**PEER REVIEW HISTORY (review-history.pdf).** An accounting of the reviewer comments and feedback.

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
