## [Reviewer comments · mSystems]

Optimizing fungal DNA extraction and purification for Oxford Nanopore untargeted shotgun metagenomic sequencing from simulated hemoculture specimens

Nattapong Langsiri, Wieland Meyer, László Irinyi, Navaporn worasilchai, Nuttapon Pombubpa, Thidathip Wongsurawat, Piroon Jenjaroenpun, Jennifer LUANGSA-ARD, and Ariya Chindamporn

Corresponding Author(s): Ariya Chindamporn, Chulalongkorn University

Review Timeline:

Submission Date:	August 29, 2024
Editorial Decision:	October 25, 2024
Revision Received:	December 20, 2024
Accepted:	March 6, 2025

Editor: Benjamin Wolfe

Reviewer(s): The reviewers have opted to remain anonymous.

Transaction Report:

DOI: <https://doi.org/10.1128/msystems.01166-24>

Re: mSystems01166-24 (Optimizing fungal DNA extraction and purification for Oxford Nanopore untargeted shotgun metagenomic sequencing from simulated hemoculture specimens)

Dear Dr. Ariya Chindamporn:

I have had an additional reviewer assess your manuscript (see the comments at the bottom of this email). In light of their comments and your past review at JCM, I would like to offer you the chance to make modifications to your manuscript before we can further consider it at mSystems.

Please note that the original reviews at JCM and the additional reviewer noted that important details are missing from the methods for your paper. Since this is a methods manuscript, please make sure that all critical methodological details are provided so that others could replicate your study.

There are also grammar issues throughout the manuscript. Please read it carefully and be sure to make sure the writing is clear and accurate.

Please also make sure that all limitations of this method and your study are noted in the Discussion. For example, you only worked with a limited number of fungal species and strains.

In addition to addressing the reviewer comments below, be sure your revisions fully address the comments provided from the JCM review.

Revision Guidelines

Sincerely,
Benjamin Wolfe
Editor
mSystems

Reviewer #1 (Comments for the Author):

This will be an important paper as ONT sequencing is increasing being evaluated as a potential rapid microbial diagnostic tool. Understanding and development of appropriate methods of DNA extraction and library preparation will be essential if ONT sequencing is to be successfully developed as a clinical microbial diagnostic tool.

Overall, the reviewer found the manuscript well-written, and the study included the appropriate background information, methodology and analysis for the research question. However, there were some concerns.

Major concerns:

The major concern was that the manuscript is a methodology study but lacked critical technical information, particularly with regards to the ONT sequencing portion of the study.

The authors have not addressed the previous reviewer's comments regarding what DNA amount was used. Library volume (75µl) is not a measure of DNA amount. This reviewer agrees with previous reviewer that the flow cells looked underloaded. For ONT ligation sequencing 100-200 fmol of gDNA is recommended. What fmol was loaded into the flow cells and was the same amount used for all the sequence runs/ samples? Additionally, which method of DNA quantification (Nanodrop, Quantus). Nanodrop typically overestimates DNA concentration (a finding this paper demonstrates) and so would not be recommended for ONT sequencing as use of Nanodrop readings can cause underloading of the flow cell. As this is a methodology manuscript, this issue should be addressed and discussed, and the authors should include which method of DNA quantification was used to determine how much of sample was loaded into the flow cell. The authors also do not mention how much of each DNA sample underwent the ONT ligation-native barcoding protocol. The protocol recommends 100-200fmol for each step (End-prep, native barcode ligation, adaptor ligation. How much DNA used at each step should also be included. It should also be mentioned how many samples per flow cell were included and the length of the sequence run.

It's also not mentioned which basecalling model was used. minKNOW offers different basecalling models (i.e. Fast, High accuracy, Super accurate) and so the authors should also include which type of basecalling was used. The authors have also made a mistake on line 245. Adaptor ligation is performed using NEBNext Quick Ligation Module (E6056), including the NEBNext Quick Ligation Reaction Buffer (5X) and Quick T4 DNA Ligase. This should be corrected.

The reviewer is also concerned that there do not appear to be any negative controls. As this manuscript uses untargeted metagenomic sequencing, any contaminating human and/or microbial DNA would be sequenced. It is therefore critical to have negative controls to determine whether contamination occurred and whether any of the discussed protocols had an increased risk of contamination. The unclassified reads, for example, may have come from bacterial species present in the hemoculture or introduced during the sample preparation steps. The author, therefore, recommends that the authors reevaluate their interpretation of why the shorter fragments could not be classified and consider aligning the unclassified reads to a database of bacterial genomes using Kracken or Metaphlan, for example.

Minor comments:

- The authors do not mention this, but it is perhaps worth mentioning that metagenomic sequencing has the ability to detect antifungal genes, which would be of particular use in the clinical setting.
- Authors should be consistent with their figures. Figures 1-2 use "Method 1" etc whilst Figure 4 uses short descriptions. The authors should select one approach and be consistent.
- In S2, 12 hours for a phenol chloroform DNA isolation seems long. This reviewer's lab has regularly performed phenol chloroform extraction on clinical samples for detection of bacteria and/ or fungi and the protocol typically takes 4 hours or less. Has the phenol chloroform protocol been modified/ extended to improve fungal detection?

Response to Reviewer's comments

The grammatical and general changes were made as highlighted in the main text to make the revision's writing clearer and more accurate.

Major comment

Question 1

The authors have not addressed the previous reviewer's comments regarding what DNA amount was used. Library volume (75µl) is not a measure of DNA amount. This reviewer agrees with previous reviewer that the flow cells looked underloaded. For ONT ligation sequencing 100-200 fmol of gDNA is recommended. What fmol was loaded into the flow cells and was the same amount used for all the sequence runs/ samples? Additionally, which method of DNA quantification (Nanodrop, Quantus). Nanodrop typically overestimates DNA concentration (a finding this paper demonstrates) and so would not be recommended for ONT sequencing as use of Nanodrop readings can cause underloading of the flow cell. As this is a methodology manuscript, this issue should be addressed and discussed, and the authors should include which method of DNA quantification was used to determine how much of sample was loaded into the flow cell. The authors also do not mention how much of each DNA sample underwent the ONT ligation-native barcoding protocol. The protocol recommends 100-200fmol for each step (End-prep, native barcode ligation, adaptor ligation. How much DNA used at each step should also be included. It should also be mentioned how many samples per flow cell were included and the length of the sequence run.

It's also not mentioned which basecalling model was used. minKNOW offers different basecalling models (i.e. Fast, High accuracy, Super accurate) and so the authors should also include which type of basecalling was used. The authors have also made a mistake on line 245. Adaptor ligation is performed using NEBNext Quick Ligation Module (E6056), including the NEBNext Quick Ligation Reaction Buffer (5X) and Quick T4 DNA Ligase. This should be corrected.

Answer

The library preparation steps were added in detail in line 166-179 including the input DNA for the library preparation steps and flow cell loading steps and the methods for adjusting the amount of DNA, the mistake in the ligation kit has been corrected in line 173-174 and base calling step was clarified in line 183-184.

Question 2

The reviewer is also concerned that there do not appear to be any negative controls. As this manuscript uses untargeted metagenomic sequencing, any contaminating human and/or microbial DNA would be sequenced. It is therefore critical to have negative controls to determine whether contamination occurred and whether any of the discussed protocols had an increased risk of contamination. The unclassified reads, for example, may have come from bacterial species present in the hemoculture or introduced during the sample preparation steps. The author, therefore, recommends that the authors reevaluate their interpretation of why the shorter fragments could not be classified and consider aligning the unclassified reads to a database of bacterial genomes using Kracken or Metaphlan, for example.

Answer

We have expanded the section on controls and provided additional discussion in line 550-582. This revision includes details on spiking microorganisms into hemoculture, following a rigorously selected protocol established through our extraction process testing. This approach has allowed us to measure the quantity of human DNA—one of the potential sources of contamination—both with and without the human DNA depletion step. Consequently, we could assess the potential contamination levels of human DNA before and after implementing the human DNA depletion protocol, as now discussed in line 555-560.

For the unclassified reads observed, we hypothesize that these reads likely represent DNA fragments insufficient for species-level identification. However, after assembly, we found that most of these fragments could indeed be accurately identified at the species level. This suggests that these unclassified reads are unlikely to represent contaminant DNA.

Regarding bacterial DNA contamination, which we suspect may arise from processing errors, this study did not conduct specific tests to quantify it, and we have acknowledged this as a limitation in line 566-570. Nonetheless, we believe such bacterial contamination to be minimal and not statistically significant relative to the other read portions, as illustrated in Figure 7. This inference supports the validity of our results, as discussed in our revised response.

Minor comment

Question 1

The authors do not mention this, but it is perhaps worth mentioning that metagenomic sequencing has the ability to detect antifungal genes, which would be of particular use in the clinical setting.

Answer This has been added to the discussion in line 573-582

Question 2

- Authors should be consistent with their figures. Figures 1-2 use "Method 1" etc. whilst Figure 4 uses short descriptions. The authors should select one approach and be consistent.

Answer Edited as commented

Question 3

- In S2, 12 hours for a phenol chloroform DNA isolation seems long. This reviewer's lab has regularly performed phenol chloroform extraction on clinical samples for detection of bacteria and/ or fungi and the protocol typically takes 4 hours or less. Has the phenol chloroform protocol been modified/ extended to improve fungal detection?

Answer The duration of the phenol/chloroform extraction protocol has been extended due to variable times required for isopropanol incubation and DNA drying, which can range from 30 minutes to overnight. We have adjusted the table to reflect this variability in Table S2.

Re: mSystems01166-24R1 (Optimizing fungal DNA extraction and purification for Oxford Nanopore untargeted shotgun metagenomic sequencing from simulated hemoculture specimens)

Dear Dr. Ariya Chindamporn:

Your manuscript has been accepted for publication at mSystems, and I am forwarding it to the ASM production staff for publication. Your paper will first be checked to make sure all elements meet the technical requirements. ASM staff will contact you if anything needs to be revised before copyediting and production can begin. Otherwise, you will be notified when your proofs are ready to be viewed.

Sincerely,

Benjamin Wolfe
Editor
mSystems